# Effectiveness of Acceptance and Commitment Therapy (ACT) for the Management of Postsurgical Pain: Study Protocol of a Randomized Controlled Trial (SPINE-ACT Study)

**DOI:** 10.3390/jcm12124066

**Published:** 2023-06-15

**Authors:** Juan R. Castaño-Asins, Juan P. Sanabria-Mazo, Juan V. Luciano, Alberto Barceló-Soler, Luis M. Martín-López, Alejandro Del Arco-Churruca, Jesús Lafuente-Baraza, Antonio Bulbena, Víctor Pérez-Solà, Antonio Montes-Pérez

**Affiliations:** 1Mental Health Care Program, INAD, Hospital del Mar, 08003 Barcelona, Spain; jrcastano@psmar.cat; 2Teaching, Research & Innovation Unit, Parc Sanitari Sant Joan de Déu, 08830 St. Boi de Llobregat, Spain; juanpablo.sanabria@sjd.es; 3Department of Basic, Developmental and Educational Psychology, Autonomous University of Barcelona, 08193 Cerdanyola del Vallès, Spain; 4Centre of Biomedical Research in Epidemiology and Public Health (CIBERESP), 28029 Madrid, Spain; 5Department of Clinical & Health Psychology, Autonomous University of Barcelona, 08193 Cerdanyola del Vallès, Spain; 6Institute of Health Research of Aragon (IIS), 50009 Zaragoza, Spain; 7Department of Psychiatry, INAD, Hospital del Mar, 08003 Barcelona, Spain; lmmartin@psmar.cat; 8Spinal Surgery Unit, Traumatology, Hospital del Mar, 08003 Barcelona, Spain; adelarco@psmar.cat; 9Spine Surgery Unit, Neurosurgery, Hospital del Mar, 08003 Barcelona, Spain; 10Department of Psychiatry and Forensic Medicine, Autonomous University of Barcelona, 08193 Cerdanyola del Vallès, Spain; antoni.bulbena@uab.cat; 11INAD, Hospital del Mar, 08003 Barcelona, Spain; vperezsola@psmar.cat; 12Pain Clinic Unit, Anesthesiology, Hospital del Mar, 08003 Barcelona, Spain; amontes@psmar.cat

**Keywords:** low back pain, lumbar spine surgery, acceptance and commitment therapy, randomized controlled trial, study protocol

## Abstract

Research on the use of Acceptance and Commitment Therapy (ACT) for patients with degenerative lumbar pathology awaiting surgery are limited. However, there is evidence to suggest that this psychological therapy may be effective in improving pain interference, anxiety, depression, and quality of life. This is the protocol for a randomized controlled trial (RCT) to evaluate the effectiveness of ACT compared to treatment as usual (TAU) for people with degenerative lumbar pathology who are candidates for surgery in the short term. A total of 102 patients with degenerative lumbar spine pathology will be randomly assigned to TAU (control group) or ACT + TAU (intervention group). Participants will be assessed after treatment and at 3-, 6-, and 12-month follow-ups. The primary outcome will be the mean change from baseline on the Brief Pain Inventory (pain interference). Secondary outcomes will include changes in pain intensity, anxiety, depression, pain catastrophizing, fear of movement, quality of life, disability due to low back pain (LBP), pain acceptance, and psychological inflexibility. Linear mixed models will be used to analyze the data. Additionally, effect sizes and number needed to treat (NNT) will be calculated. We posit that ACT may be used to help patients cope with the stress and uncertainty associated with their condition and the surgery itself.

## 1. Introduction

Chronic pain is considered a multidimensional experience, and it is accepted that both the intensity and the characteristics of pain are influenced by the individual’s cognitive processing [1]. Chronic pain involves a high comorbidity with mental disorders and the presence of clinically relevant psychopathology [2,3]. There is evidence of the effectiveness of Acceptance and Commitment Therapy (ACT) in patients with chronic pain, both in individual and group formats [4,5,6,7,8,9].

Specifically, ACT maintains that language is at the base of psychological problems, making it inevitable that under certain conditions thoughts and sensations arise that can be experienced as annoying. The fact of being verbal also makes it easier for people to become entangled in fighting against their own private events, and to persist in them even though the results of such struggles are often counterproductive. Using metaphors, paradoxes, and experiential exercises, people learn to be in touch with thoughts, feelings, memories, and sensations, both those previously feared and avoided and any others that arise. In this way, people learn the ability to recontextualize these private events, clarify what matters to them in their lives—what deep down and radically they value—and gain commitment to the necessary changes in the action [10,11].

Low back pain (LBP) is a problem that affects around 11 to 84% of the general population at some point in their lives [12,13]. There are several environmental and personal factors that influence the onset and course of LBP, including comorbid anxiety and depression, as well as certain types of cognitive variables, such as pain catastrophizing [14]. Particularly, pain catastrophizing is one of the psychological constructs that has been most researched in relation to pathologies that cause pain. It is defined as a set of cognitive and emotional processes that predispose pain to become chronic. Individuals who catastrophize develop a very negative view of their pain, think about it often, and feel unable to control it, so they have a worse prognosis for any treatment [15]. Therefore, early detection of risk factors for LBP has been identified as a strategy for identifying patients who may be at risk of poor clinical outcomes, and as a potential method for improving the effectiveness and cost-effectiveness of psychological therapies [16].

Degenerative low back conditions, such as spinal stenosis and facet arthrosis, lead to chronic pain, impairment, and reduced quality of life [17]. The prevalence in the general population varies from 20% to 25% and increases to over 45% in people over 60 years of age [18,19,20]. Lumbar spinal stenosis is one of the most common diagnoses associated with spinal surgery [21,22]. Surgical treatment for lumbar degenerative conditions is well established and studies have reported that the benefits of surgery are maintained in the long term, although there are hardly any studies that compare it with other types of treatments such as pharmacological ones [23]. Despite advances in surgery, after lumbar spine surgery, adults continue to have poorer physical conditions and mental health outcomes compared to the general population [24,25]. Studies have found persistence of pain, functional disability, and poor quality of life in up to 40% of people after spinal surgery for lumbar degenerative conditions [26,27,28,29]. The reoperation rate has been reported to range from 18% to 23% between 8 and 10 years after surgery [30]. Even though there are logical benefits of surgical treatments, there is also a risk that the patient develops so-called chronic postsurgical pain (CPSP), the incidence of which varies depending on the type of operation between 5% and 85% [31].

According to the International Association for the Study of Pain (IASP), CPSP is defined as chronic pain that develops or increases in intensity after a surgical procedure and is categorized as secondary chronic pain [32]. Although the risk factors for CPSP are common with other types of chronic pain, current evidence has established that the factor most associated with this pathology is the duration of intense acute postoperative pain [33]. This kind of pain can generate central sensitization, which lowers the mechanical threshold and exaggerates the response to noxious stimuli. Therefore, the patient may present with both hyperalgesia and allodynia. Reports indicate that between 10% and 70% of patients requiring major surgery go on to develop CPSP depending on the type of surgery (e.g., cardiac, thoracic, gynecologic, and spinal) [34]. A recent meta-analysis [35] found an association between anxiety, catastrophizing, depression, kinesiophobia, and CPSP. People with chronic degenerative LBP who are candidates for surgery, with higher rates of catastrophizing, anxiety, depression, and fear of movement (kinesiophobia), will have a higher risk of pain chronification and worse prognosis in relation to the efficacy of surgical treatment.

The first step in addressing this type of pain, as well as the disability and suffering it entails, is to identify the modifiable and protective causal risk factors that predict the CPSP. The identification of psychological risk and protective factors for CPSP will serve for the development and implementation of psychological interventions to prevent and/or manage CPSP [35]. Expanding knowledge of the psychological and social predictors of CPSP is essential to recognize patients at risk of poor outcomes. As psychological risk factors for CPSP, anxiety (both as a state and as a trait), depression, catastrophizing, and kinesiophobia have a significant association with chronic postoperative pain. Of the psychological predictors that have a significant association with chronic postoperative pain, the state of anxiety is the most explanatory. Anxiety is understood as an emotional reaction or as a personality trait [35,36]. Trait anxiety refers to the individual tendency to react anxiously, while the state is described as a transient emotional state that fluctuates over time. Kinesiophobia, defined as “an excessive, irrational, and debilitating fear of physical movement and activity resulting from a feeling of vulnerability due to painful injury or reinjury” is found to be a central factor in the process of pain developing from acute to chronic stages [36].

There is a growing understanding of the important role that psychological interventions can play in reducing negative affect, avoidance behaviors, and pain perception after surgery. Specifically, ACT allows patients to learn to expand their ability to experience and accept pain, including the negative thoughts and feelings that inevitably arise when pain is present. Physical pain sensations, as well as patient psychological reactions to them, are observed in a neutral, non-reactive manner, while focusing on improving motivation and commitment to engage in personally meaningful, achievable, and goal-oriented activities. This is practiced without engaging in problematic avoidance behaviors, which typically worsen pain and limit functioning. Patients become more psychologically flexible and learn to adapt their behavior in a way that allows them to live a rich and meaningful life [37]. Finally, previous studies have shown that psychological predictors have a significant association with the development of CPSP. Kinesiophobia and depression are independently associated with chronic postsurgical pain and disability and decreased physical function after lumbar spine surgery [38,39,40,41,42,43,44].

The present work describes the design of a randomized controlled trial (RCT) comparing the effectiveness of preoperative ACT versus treatment as usual (TAU) to improve pain interference in patients subjected to surgery for degenerative lumbar pathology. Considering the above evidence, we hypothesize that participants who engage in ACT will have greater improvement in quality of life and less disability and pain interference compared to participants who do not engage in ACT. The objectives of this RCT are the following: (a) to analyze the effectiveness of ACT as an adjuvant to TAU to improve pain interference (primary outcome); (b) to examine the effectiveness of ACT for improving pain intensity, anxiety, depression, pain intensity, quality of life, functional status, kinesiophobia, pain catastrophizing, psychological flexibility, and pain acceptance (secondary outcomes); and (c) to explore the differences between responders and non-responders in terms of sociodemographic and clinical characteristics.

## 2. Materials and Methods

### 2.1. Design

This study is a 12-month RCT that follows the guidelines of the Standard Protocol Items: Recommendations for Interventional Trials (SPIRIT) [45] and the Consolidated Standards of Reporting Trials (CONSORT) [46]. The study has two treatment arms: a control group that receives TAU and an intervention group that receives TAU + ACT. The study is registered on ClinicalTrials.gov, accessed on 5 December 2022, with the NCT number NCT05634122. Patients in both arms will receive TAU as it is commonly provided by the Spanish National Health System [47].

### 2.2. Participants and Sample Size

The sample size was estimated through R with R Studio using the package “pwr.t.test” version 4.3.0. This sample size was determined to achieve a power of 80%, a significance level of 5%, and a treatment effect size based on Cohen’s *d* of 0.64 and 0.73 for pain interference (BDI) at post-treatment and at follow-up, respectively [48]. Considering a dropout rate of 30% [48,49,50,51], it will be necessary to include 51 subjects in each study arm. A total of 102 patients with degenerative lumbar pathology who are candidates for lumbar surgery from the Spine Unit of Hospital del Mar (Barcelona, Spain) will be recruited for this study.

Patients who have been scheduled for surgery in the Spine Unit and who have been diagnosed with degenerative pathology that also present psychosocial risk factors for CPSP, as determined by screening questionnaires, will be assessed by the Pain Unit psychiatrist starting June 2023 and will be recruited if they meet the selection criteria. Inclusion criteria will be (a) age 18–80 years, (b) adequate understanding of Spanish or Catalan, (c) meeting the diagnostic criteria for the LBP/degenerative low back pathology that indicates a candidate for first surgical treatment according to the clinical history (International Classification of Diseases [ICD]-10 codes for surgical and diagnostic processes) [52], (d) presence of psychosocial risk factors for CPSP, (e) having an internet connection on a smartphone, tablet, or computer to be able to participate in online therapy, and (f) giving authorization by signing informed consent. Specifically, the psychosocial risk factors associated with CPSP are anxiety, depression, catastrophizing, and fear of movement [35,36]. During the outpatient visit to the Spine Unit, and once the surgical intervention has been proposed and accepted by the patient, the patient will complete three self-administered questionnaires: The Hospital Anxiety and Depression Scale (HADS), Pain Catastrophism Scale (PCS), and Tampa Scale of Kinesiophobia (TSK-11). To select the participants, the cut-off points are high in relation to the HADS (cut-off point ≥ 11), PCS (cut-off point ≥ 24), and TSK-11 (cut-off point ≥ 27), which are based on what was suggested by the authors and reviewers of the adaptations of each questionnaire [53,54,55].

Exclusion criteria will be (a) cognitive impairment (understood as a decline in a person’s cognitive abilities, such as memory, attention, language, and reasoning,) that, in the clinician’s opinion, prevents them from participating in therapy, (b) presence of chronic pain not related to the diagnosis of LBP that indicates a candidate for surgical treatment, (c) not presenting psychosocial risk factors for CPSP, (d) being involved in litigation or pending legal claims for compensation, (e) having a severe or decompensated psychiatric disorder (such as severe depression, schizophrenia, bipolar disorder, or personality disorder) that, in the clinician’s opinion, prevents participation in therapy, (f) suicidal ideation, (g) consumption of toxic substances in abuse/dependence, except for smoking, and (h) having a surgical indication for lumbar surgery that involves reintervention.

### 2.3. Procedure

The principal investigator (J.R.C.-A.), through an initial interview, will provide a general description of the study to patients with an indication for lumbar surgery interested in participating who met the eligibility criteria. Prior to random assignment to treatment (TAU or TAU + ACT), informed consent forms will be collected, which will include a general description of the characteristics of the interventions. Patients will also be informed that their participation will be voluntary and that they may withdraw at any time, with the guarantee that they will continue to receive their TAU. ACT will be performed by the psychiatrist (J.R.C.-A.) who is trained and accredited in ACT linked to the hospital’s Pain Unit, who has years of experience in individual and group management with this therapy at the level of patients with psychiatric comorbidity and chronic pain. The psychological intervention will be delivered before the participants have undergone surgery. At present, for our current protocol, in the context of the heterogeneity of the applications of perioperative psychological therapies in the studies published to date, it is not clear if patients who are candidates for surgery can benefit more from psychotherapeutic interventions in the preoperative period, in the postoperative period, or combined. In addition, the exact number of sessions and the adequate time to obtain improvement is unknown [56]. In our center, we have experience in patients with chronic pain with the 8-session ACT program. Given that the patients candidates for the study of our sample are a population at high risk of CPSP (i.e., already with preoperative pain) and considering the logistics that we can apply in our hospital (i.e., material and human resources), we find it interesting to study the response of 8 complete ACT therapy sessions in the preoperative period of this population and explore and share the findings.

This research will be carried out in accordance with the ethical standards established in the Declaration of Helsinki of 1964 and was approved by the Ethics Committee of the hospital (PR(ID) 2021/9998/I). Patient data will be treated confidentially, ensuring that only the research team can access this information after recoding the name and personal identity number. Only the principal investigator (J.R.C.-A.) will have access to the patient’s code key, which, in accordance with current data protection legislation in Spain, will be kept separately in a secure place. Participants will be assigned to the intervention group (TAU + ACT) or control group (TAU) using an SPSS v25 randomization list. Patients will be evaluated through 5 visits: 1 baseline visit (Visit 1), another visit after the ACT intervention (Visit 2), and other visits at 3, 6, and 12 months after surgery (Visits 3, 4, and 5). The RCT flowchart is presented in Figure 1.

#### Forecast Execution Dates

Inititation of patient recruitment: June 2023;End of patient follow-up period: December 2024;Publication of results: June 2025.

### 2.4. Treatments

#### 2.4.1. Intervention Group (TAU + ACT)

ACT is a new form of cognitive behavioral therapy (CBT) with a specific focus on mindfulness and acceptance of difficult emotional experiences. It is an evidence-based behavioral therapy that has been found to be effective in treating chronic pain [57]. This therapy emphasizes the need to accept aversive thoughts and feelings, rather than trying to control them. It also promotes a commitment to acting towards personal values, which can lead to a more meaningful and fulfilling life. This is achieved by developing “psychological flexibility” through key processes, such as acceptance, differentiation of self from aversive thoughts and emotions, cognitive defusion (i.e., the ability to observe one’s own thoughts and sensations as temporary events, without necessarily considering them to be true or reflecting oneself), identifying and taking actions based on personal values, being in the present moment, developing an “observing self”, and taking committed action that aligns with personal values. There is evidence that trying to control or change pain can be counterproductive and can lead to increased distress and decreased functionality and quality of life [58,59]. ACT can be an effective treatment for promoting psychological acceptance and flexibility in relation to chronic pain, which can be challenging to modify [60]. The ACT program will be conducted at the Hospital del Mar Pain Unit online (preferably) or in person and will consist of 8 sessions, held once a week, each lasting 90 min. The therapy will last 8 weeks, and each group will have a maximum of 15 participants. The principal investigator (J.R.C.-A.), a psychiatrist with specialization in ACT, will oversee delivering the treatment sessions. ACT will be based on the Vowles et al. protocol [61]. To monitor treatment within ACT, the sessions will be video-taped. An expert in ACT will rate adherence to treatment by revising the videotapes of therapy sessions. A random sample of tapes will be rated using the Acceptance and Commitment Therapy Fidelity Measure. The ACT-FM is a 25-item measure that captures 4 areas: therapist stance, open response style, aware response style, and engaged response style (each split into ACT consistent and ACT inconsistent items, making 8 sections in all). Items are rated on a 4-point scale from 0 (“behavior never occurred”) to 3 (“therapist consistently enacts this behavior”) [62].

Table 1 below shows the contents of the 8 sessions (Session 1: Presentation of the general ACT concept; Session 2: Value analysis I; Session 3: Value analysis II; Session 4: Value analysis III; Session 5: Values and feelings; Session 6: Drink one direction; Session 7: Dare and change; Session 8: Ready to act with ACT).

#### 2.4.2. Control Group (TAU)

The control group (TAU) in this study primarily consists of medication prescription and routine follow-up according to the symptoms of each patient. Patients in the control group may also receive ACT treatment if deemed appropriate and recommended by healthcare providers after the study is finished, if they so choose.

### 2.5. Study Measures

Participants in the intervention group (TAU + ACT) and the control group (TAU) will complete the instruments described below. The Table 2 shows the time points at which study measures will be administered.

#### 2.5.1. Measures

This will be used to gather general and clinical information about the patient, including age, educational level, marital status, and years since the diagnosis of LBP, among others.

##### Clinical Features and Screening Measures

The baseline visit of the study will include a clinical examination by a psychiatrist (J.R.C.-A.) from the Pain Unit of Hospital del Mar (Barcelona, Spain) to diagnose mental disorders, using ICD-10 diagnostic criteria and following the usual protocol of the Pain Unit psychiatrists. The examination of the patients by the orthopedic specialist and neurosurgeons of the Spine Unit of the Hospital del Mar will be used to diagnose disorders at the level of the lumbar spine and the type of surgical intervention, using ICD-10 diagnostic criteria, which will be recorded in our protocol.

##### Primary Outcome Measure

The Brief Pain Inventory (BPI) [63,64] is a self-report measure that assesses chronic pain in a multidimensional way. It provides information on pain interference (primary outcome) and pain intensity (secondary outcome) with patients’ daily functioning and activities. An abbreviated version of 11 items is available. The first 4 items evaluate the pain intensity in the last 24 h, ranging from 0 (no pain) to 10 (the worst pain imaginable). Subsequently, 7 items (i.e., general activity, mood, ability to walk, ability to work including domestic work, relationships with other people, sleep, and ability to enjoy the life) are assessed from 0 (does not disturb) to 10 (totally disturbs) to report the degree of involvement of chronic pain in these areas of the person’s functioning. The higher the score in the areas of the questionnaire, the greater the intensity of perceived pain, as well as the greater degree of involvement of chronic pain in the functioning of the person in the various areas explored.

##### Secondary Outcome Measures

The Hospital Anxiety and Depression Scale (HADS) [65,66] is a 14-item questionnaire, consisting of two 7-item sub-scales, one for anxiety (HADS-Anx, odd items) and another for depression (HADS-Dep, even items). Symptoms are evaluated on a 4-point Likert scale (range 0–3) with different response options. The time frame is within the previous week. The score for each sub-scale is obtained by adding the values of the respective items. The same cut-off points are proposed for the two sub-scales (0–7 normal; 8–10 possible involvement; ≥ 11 clinical problem).

The Pain Catastrophizing Scale (PCS) [67] is a scale with 13 items that includes 3 dimensions: rumination (items 8, 9, 10, and 11), magnification (items 6, 7, and 13) and helplessness (items 1, 2, 3, 4, 5, and 12). Each item is scored from 0 (not at all) to 4 (always) and the total scores range from 0 to 52, with higher scores indicating greater pain catastrophizing.

The Tampa Scale of Kinesiophobia (TSK-11SV) [68,69] is a questionnaire that allows us to measure the fear of moving or being injured/(re)injured in patients with pain. There is a reduced version that is validated in Spanish and is called TSK-11SV. This scale consists of a questionnaire, which contains 11 items in its reduced version. Scores range from one to four points with a response of “strongly disagree”, meaning one point; “partially disagree”, two points; “partially agree”, three points; and “strongly agree”, 4 points. To obtain a total final score, the sum of the items is required. The final score can be at least 11 and at most 44 points. The higher the score, the higher the degree of kinesiophobia.

The Chronic Pain Acceptance Questionnaire (CPAQ) [70,71] consists of 20 items that assess 2 main factors: “willingness or involvement in activities” (activity engagement) and “acceptance or openness to pain” (pain willingness). Respondents rate each of the 20 items on a scale from 0 (never true) to 6 (always true). Nine of these items measure acceptance or openness to pain (items 4, 7, 11, 13, 14, 16, 17, 18, and 20), and eleven measure willingness or involvement in activities (items 1, 2, 3, 5, 6, 8, 9, 10, 12, 15, and 19). The total score of the questionnaire is the sum of the direct items of involvement in activity engagement, plus the sum of the items measuring pain willingness. The maximum possible total score is 120, with higher scores indicating greater pain acceptance.

The Psychological Inflexibility of Pain Scale (PIPS) [72,73,74] consists of 12 items that assess 2 main factors, known as avoidance and cognitive fusion, on a 7-point Likert-type scale ranging from “never” to “always true”. Higher PIPS scores indicate lower psychological flexibility.

The SF-12 Health Questionnaire (SF-12) [75] is composed of a subset of 12 items from the SF-36, from which the physical and mental summary components of the SF-12 are constructed as single scores. There are 12 items, with 8 dimensions and 2 summative components (physical and mental). The response options form Likert-type scales that assess the intensity or frequency. The number of response options ranges from three to six, depending on the item. Values higher or lower than 50 should be interpreted as better or worse, respectively, than the reference population. For each of the 8 dimensions, the items are coded, aggregated, and transformed into a scale that ranges from 0 (the worst state of health for that dimension) to 100 (the best state of health).

The Oswestry Low Back Pain Disability Scale (OLBPDQ) [76,77] is a questionnaire specific for LBP that measures limitations in daily activities. It consists of 10 questions with 6 possible answers each. The first question refers to the intensity of the pain, specifying in the different options the response to taking analgesics. The remaining items include basic activities of daily living that can be affected by pain (i.e., self-care, lifting weights, walking, sitting, standing, sleeping, sexual activity, social life, and travel). The scale has 10 questions with 6 possible answers each. Each item is valued from 0 to 5, from least to most limiting. If the first option is marked, 0 is scored and 5 if the last option is indicated. The total score, expressed as a percentage (from 0 to 100%), is obtained by summing the scores for each item, divided by the maximum possible score multiplied by 100. Higher values describe greater functional limitation. Score ranges are as follows: 0–20%, minimal functional limitation; 20–40%, moderate; 40–60%, intense; 60–80%, disability; and above 80%, maximum functional limitation.

### 2.6. Statistical Analysis

Data analyses will be carried out using the statistical packages SPSS v26.0, R with R Studio v4.0.2, and MPlus v7.4. Comparative analyses of continuous data will be performed using the Student’s *t* test or their non-parametric equivalents (i.e., Mann–Whitney U or Kruskal–Wallis test), depending on the characteristics inherent to the variables under study. The Chi-Square test or Fisher’s exact test will be used, as appropriate, for categorical variables. The Bonferroni correction will be applied in all post hoc tests. In all statistical tests performed with the outcome variables, a statistical significance level of 0.05 will be used. In adherence with CONSORT recommendations, potential baseline differences in sociodemographic and clinical characteristics are considered irrelevant after randomization and, as a result, were not included as covariates in the analyses of RCT outcomes.

The main analysis will compare the effect ACT compared to TAU on the primary outcome (pain intensity and pain interference at 12-month follow-ups). All data analyses will be carried out following an intention-to-treat (ITT) principle, that is, regardless of protocol adherence. Then, we will compute analysis of the primary outcome post-treatment and analysis of the secondary and treatment process outcomes at post-treatment and at 12-month follow-up. The analyses will be replicated from a per-protocol approach. Multi-level, linear mixed models will be created using the restricted maximum likelihood method for the estimation of parameters. The effect sizes will be calculated according to Cohen’s *d*. A 5% significance level will be used in all 2-tailed tests, applying the Benjamini–Hochberg correction for multiple comparisons (to reduce the risk of false positives).

To determine the clinical significance of improvements on the primary outcome (BPI), patients will be classified into two categories: responders and non-responders. Considering the Initiative on Methods, Measurement, and Pain Assessment in Clinical Trials (IMMPACT) recommendations, a 1-point reduction in the pre–post and pre-follow-up BPI total score are used as a criterion for establishing clinically significant improvement [78]. This categorization was also used to calculate the number needed to treat (NNT) in TAU + ACT compared to TAU, with a 95% confidence interval (CI) calculated for each NNT at post-treatment and at follow-up. Furthermore, between-group differences in sociodemographic and clinical characteristics, and outcomes at baseline, post-treatment, and follow-up will be examined for responders versus non-responders using the Chi-Square test or Fisher’s exact test and Student’s *t*-tests for categorical and continuous variables, respectively.

## 3. Discussion

The present work describes the design and protocol of an RCT that aims to assess the effectiveness of ACT compared to TAU in patients with degenerative pathology of the lumbar spine who are candidates for surgery. The protocol for this RCT was designed using SPIRIT recommendations and registered in a clinical trial database (ClinicalTrials.gov, accessed on 5 December 2022) according to CONSORT guidelines. The interest is to explore the benefits of ACT applied before surgery in degenerative lumbar pathology to improve the results in pain and functionality in patients with a high risk of developing chronic pain.

Based on previous work from RCTs with ACT applied perioperatively in orthopedic and breast surgery [79,80], in which a decrease in pain interference and an improvement in anxiety levels were observed, we expect to be able to provide the scientific community with greater evidence of the usefulness of ACT perioperatively in relation to the pain interference in activities of daily living and the psychological variables explored in our study.

If the RCT findings are robust enough, ACT could be offered as an additional intervention to TAU, with a coherent strategic approach that is adjustable to the health resources allocated to patients who are candidates for lumbar surgery. In this RCT, in addition to evaluating the clinical effects of ACT in the short, medium, and long term, we will seek to recognize relevant moderators and mediators of clinical change. The possible withdrawal of patients from the trial will be one of the risk components that will be considered. In this sense, a sensitivity analysis will be developed as a strategy to determine the impact of adherence to the ACT protocol on the observed effects.

Within the process of community care and special programs of the Insititut de Neuropsiquiatria i Addiccions (INAD) of the PSMAR, there is a long tradition of care in the use of different group therapy programs in the adult population receiving inpatient and outpatient care. As part of this process, psychological group therapies are carried out in the hospital setting of the PSMar Pain Unit, which are aimed at improving coping with chronic pain by applying a therapeutic protocol based on ACT. We have adapted our ACT protocol that we use for patients with chronic pain to a pre-surgical protocol of 8 sessions in a group format, encompassing both a behavioral psychotherapeutic intervention and pain psychoeducation. Psychological intervention covers many key areas, including identifying personal functioning goals, observing, and describing physical pain and the difficult thoughts and feelings that come with pain, identifying avoidance behaviors and analyzing when they exacerbate pain, distress, anxiety, and dysfunction and noting the impact on pain of engaging in worthwhile activities gradually, with acceptance in mind.

Predicting prognosis in patients scheduled for surgery for LBP, by means of pre-assessment tests, could be efficient to apply in complementary treatment modalities, and thus prevent chronification of pain and worsening of functionality. Therefore, the objective of our study will be to test ACT in an RCT to better understand its impact in the short medium, and long term on the pain intensity, pain interference in activities of daily living, quality of life, mood, and pain disability. We will explore if applying ACT with an online preoperative program of 8 sessions, lasting 90 min, is effective, as well as analyzing patient adherence. We will investigate whether the results underscore and corroborate the possible benefits of integrating psychology services in multidisciplinary teams to prevent the incidence, intensity, and disability of CPSP. In our study, we will stratify the patients who have undergone ACT based on the number and type of psychological factors of CPSP presented, to see if there are significant differences.

It is hoped that ACT will improve the results in variables studied in relation to chronic pain in this population at high risk of postsurgical LBP. This study is justified because of the importance of identifying these psychosocial prognostic factors, and to be able to intervene with them, to improve the clinical outcomes of surgical treatment in patients with chronic degenerative LBP. The purpose is to apply ACT to improve these outcomes. Of the patients who are candidates for lumbar surgery, those with a high fear of movement (kinesiophobia), a high level of catastrophism, and the presence of anxiety and depression will be screened to focus the intervention on them. The ACT program has been designed to develop skills and tools with patients to enable them to regulate themselves in the face of pain, fear of movement, catastrophizing, anxiety, and depression. By applying this intervention, we will seek to improve pain acceptance and psychological flexibility with the idea of improving catastrophizing, kinesiophobia, anxiety, depression, quality of life, and overall personal functioning of people with chronic pain both before and after surgery, as measured by the degree of pain interference (primary endpoint of our study).

Predicting the prognosis in patients scheduled for surgery for LBP, through previous evaluation tests, could be efficient to apply them to complementary treatment modalities and thus prevent the chronification of pain and the worsening of functionality. People with chronic degenerative LBP who are candidates for surgery, with higher rates of catastrophizing, anxiety, depression, and fear of movement (kinesiophobia) will present a higher risk of chronic pain and a worse prognosis in relation to the efficacy of surgical treatment. We posit that participants receiving ACT will have greater improvement in pain intensity, pain interference, and quality of life, and less disability than participants receiving only TAU. It is hoped that ACT can improve the results in the variables studied in relation to chronic pain in this population with a high postsurgical risk of LBP.

Finally, we would like to mention that including TAU alongside an intervention in a clinical trial is not only important from a scientific perspective, but also from an ethical one. It is necessary to consider the patient’s perspective and ensure that they receive the appropriate treatment and the best possible care. In some cases, TAU may be the best available treatment for a given condition. In addition, this helps to prevent patients in the clinical trial from being subjected to unnecessary risk by receiving only the experimental intervention without having the standard treatment option.

Nevertheless, including TAU alongside an ACT in an RCT may introduce different types of biases that could affect the interpretation of the results. Confounding bias can make it difficult to separate the effects of treatment as usual from the effects of the experimental treatment. To minimize the biases that may arise when including TAU alongside an intervention in our RTC, we will carefully monitor variables that could explain the study results, such as patient characteristics (medical history, comorbidities, and medication) and intervention characteristics (adherence to treatment as usual, dropouts, administration format, therapist experience, and patient expectations). Monitoring these variables (which are collected in the RCT battery of measures) can help us to improve the validity of the study results.

## 4. Conclusions

This study represents the attempt to combine psychological therapy with lumbar spine surgery for degenerative pathology:A two-arm RCT is planned to address the potential effectiveness of ACT compared to TAU.ACT postulates the need to abandon efforts to control aversive thoughts or feelings and accept them as they are. It also promotes a commitment in a valuable vital direction for the person in relation to detecting and promoting actions that bring them closer to the important things in their life; this is useful in a life with meaning.It is hoped that ACT can improve the results in the various variables studied in relation to chronic pain in this population with a high postsurgical risk of LBP.

## Figures and Tables

**Figure 1 jcm-12-04066-f001:**
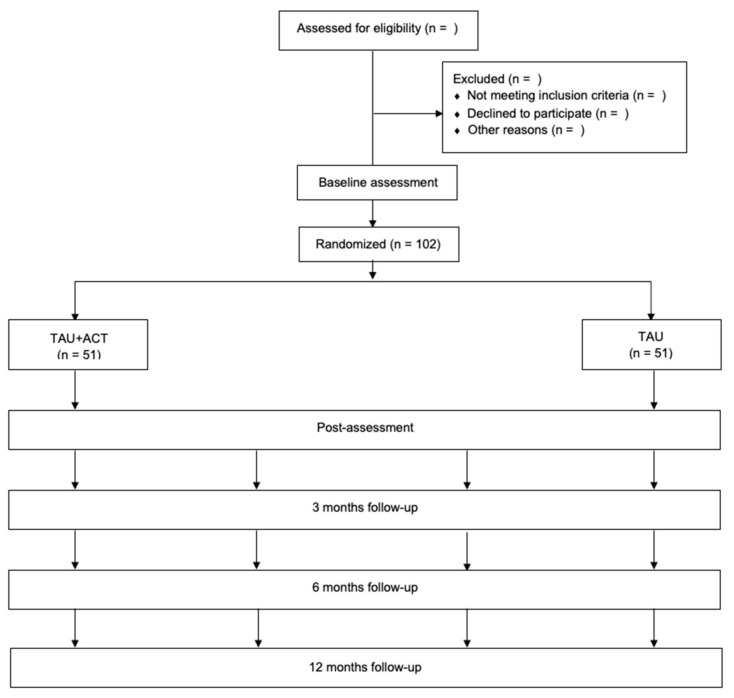
Flow chart for the randomized controlled trial (RCT).

**Table 1 jcm-12-04066-t001:** Outline of ACT group treatment sessions.

Session	ACT
1	Participants’ and clinician’s presentation. Psychoeducation and introduction to ACT (basics; scientific advances in chronic pain and depression management; psychological theories of pain, suffering, and stress; stressors, fears, and indicators; identification of values; and breathing exercises).
2	Value analysis I. Problems of experiential avoidance. Creative hopelessness through metaphors: control is the problem and not the solution. Anxiety, fight, and flight, and its effects. Accepting the risk of the life’s journey: experiences, feelings, and emotions.
3	Value analysis II. Objectives. Laws of thought and consequences of language. Mind and deactivation of thought (cognitive defusion): creating distance with thoughts. Learning meditation techniques and effects. Practicing meditation exercises.
4	Value analysis III. Psychological barriers and obstacles. Emotional distress and its consequences. Emotional phenomena, personality variables, and health states. Discovering commitments with committed actions.
5	Values and feelings. Taking the initiative with a “Plan of action and willingness”. Psychological flexibility, resilience, and self-motivation. Expansion and body scan exercises. Learning to relax.
6	Taking a direction. The self as context, process, and content. Awareness of the present: “here and now”. The brain and emotions: managing situations and overwhelming emotional responses.
7	Dare and change. Willingness and determination. Self-awareness, assertiveness, and self-esteem. Experiential expansion exercises; felt sensations. Happiness according to positive psychology. Benefits of physical exercise: movement.
8	Ready to take action with ACT. Final reflections and review of what was seen in the previous sessions. How can we apply what we have worked on, oriented to spinal surgery? Farewell and Thanks.

Note. At the beginning of each session, time will be taken to briefly go over what was discussed in the previous session and every person’s weekly records will be collected and briefly commented on. ACT: Acceptance and Commitment Therapy.

**Table 2 jcm-12-04066-t002:** Time points at which measures and data are collected.

Measures	Pre	Post	3-Months	6-Months	12-Months
Sociodemographic, clinical, and screening measures					
Sociodemographic and clinical questionnaire	X				
Primary outcome measure					
BPI (pain interference)	X	X	X	X	X
Secondary outcome measures					
BPI (pain intensity)	X	X	X	X	X
HADS (anxiety and depression)	X	X	X	X	X
PCS (pain catastrophizing)	X	X	X	X	X
PIPS (psychological inflexibility to pain)	X	X	X	X	X
TSK-11SV (fear of movement)	X	X	X	X	X
CPAQ-20 (pain acceptance)	X	X	X	X	X
SF-12 (health-related quality life)	X	X	X	X	X
OLBPDQ (functional status)	X	X	X	X	X

Note: BPI = Brief Pain Inventory-Interference Scale; HADS = Hospital Anxiety and Depression Scale; PCS = Pain Catastrophizing Scale; PIPS = Psychological Inflexibility in Pain Scale; TSK-11SV = Tampa Kinesiophobia Scale; CPAQ-20 = Chronic Pain Acceptance Questionnaire; SF-12 = Short-Form 12 Health Survey; OLBPDQ = Oswestry Low Back Pain Disability Scale.

## Data Availability

Not applicable.

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
