# Peer review of "Effectiveness of Acceptance and Commitment Therapy (ACT) for the Management of Postsurgical Pain: Study Protocol of a Randomized Controlled Trial (SPINE-ACT Study)"

_jcm, 2023, doi:10.3390/jcm12124066_

Round 1

Reviewer 1 Report (Previous Reviewer 2)

Many of the authors' edits strengthened this protocol paper. Yet, there are still several areas to address. Overall, proofreading is needed. The authors did not address all suggestions regarding grammatical errors (e.g., in the Abstract: “...for patients' candidates for surgery..." does not make sense; in the Intro: “...benefits of surgery compared to non-surgical...”). Some of the wording is also a bit unclear. For example, "...we hypothesize that participants who perform ACT therapy will have...". Patients do not "perform" ACT but rather "engage" in treatment. The additions to the intro provide helpful background, however, the intro could be better organized and more concise (e.g., describing ACT twice, in 2 different places, seems redundant). It is unclear why pain catastrophizing is discussed so early on when other psychological outcomes (e.g., anxiety, kinesiophobia) are discussed later. Also, the hypotheses could be moved to the end after the authors list the study aims. In the study aims, the authors list "pain intensity" twice (as both a primary and secondary outcome). The final sentence of the intro seems irrelevant now that the authors removed mediation/moderation analyses. In the discussion, the authors note "We hope and trust that our study will allow further research into the efficacy of psychological interventions to reduce chronic post-surgical pain and disability, with a 1-year follow-up, as well as whether there is a correlation with psychological variables associated with CPSP." However, based on study aims and proposed analyses, the authors do not plan to examine the relationships between psychological variables and CPSP.  The authors also state in the discussion, "...as well as if there is any type of patient who may benefit more from the psychological intervention, depending on their psychopathological profile." However, the authors omitted moderation analyses from their analysis plan, and thus, won't be examining who benefits most. The authors did, however, mention examining and comparing "responders vs. non-responders" in their analysis plan, but this was not noted in any of their aims. The discussion still lacks mention of prior work in this area (e.g., other studies examining perioperative ACT/psychological intervention to improve outcomes in spine surgery patients). 

Proofread and edit for grammar and clarity.

Author Response

Many of the authors' edits strengthened this protocol paper. Yet, there are still several areas to address. Overall, proofreading is needed. The authors did not address all suggestions regarding grammatical errors (e.g., in the Abstract: “...for patients' candidates for surgery..." does not make sense; in the Intro: “...benefits of surgery compared to non-surgical...”). Some of the wording is also a bit unclear. For example, "...we hypothesize that participants who perform ACT therapy will have...". Patients do not "perform" ACT but rather "engage" in treatment. The additions to the intro provide helpful background, however, the intro could be better organized and more concise (e.g., describing ACT twice, in 2 different places, seems redundant). It is unclear why pain catastrophizing is discussed so early on when other psychological outcomes (e.g., anxiety, kinesiophobia) are discussed later. Also, the hypotheses could be moved to the end after the authors list the study aims.

Following your recommendations, the wording of the complete manuscript has been reviewed and the grammatical errors have been corrected. In relation to the psychological constructs, pain catastrophizing is presented first to highlight its relevance in relation to chronic pain in general, while the other constructs are subsequently presented.

In the study aims, the authors list "pain intensity" twice (as both a primary and secondary outcome).

Both pain intensity and pain interference are measured with the BPI, but the primary outcome is pain interference. Pain intensity is a secondary outcome. For clarity purposes, we have rewritten the objectives, study measures, and Table 2. Now it’s clearer for the reader which is the primary outcome.

The final sentence of the intro seems irrelevant now that the authors removed mediation/moderation analyses. In the discussion, the authors note "We hope and trust that our study will allow further research into the efficacy of psychological interventions to reduce chronic post-surgical pain and disability, with a 1-year follow-up, as well as whether there is a correlation with psychological variables associated with CPSP." However, based on study aims and proposed analyses, the authors do not plan to examine the relationships between psychological variables and CPSP.  The authors also state in the discussion, "...as well as if there is any type of patient who may benefit more from the psychological intervention, depending on their psychopathological profile". However, the authors omitted moderation analyses from their analysis plan, and thus, won't be examining who benefits most. The authors did, however, mention examining and comparing "responders vs. non-responders" in their analysis plan, but this was not noted in any of their aims.

Fixed. We have deleted the last sentence of the "Introduction" section to be consistent with our final analysis plan. Additionally, the discussion section has been revised to make it coherent with our analysis plan. In response to your request, we have added as the third objective of this RCT “to explore the differences between responders and non-responders in terms sociodemographic and clinical characteristics". Thank you for your insightful comment.

The discussion still lacks mention of prior work in this area (e.g., other studies examining perioperative ACT/psychological intervention).

We completely agree. Therefore, we have included the following sentence in the "Discussion" section: “Based on previous RCTs with ACT applied perioperatively in patients undergoing orthopedic and breast surgery [80,81], in which a decrease in pain interference and an improvement in anxiety were observed, we expect to be able to provide the scientific community with greater evidence of the usefulness of ACT perioperatively for pain interference reduction in activities of daily”. These are the new references 80 and 81:

  • Anthony CA, Rojas EO, Keffala V, Glass NA, Shah AS, Miller BJ, HogueM, Willey MC, Karam M, Marsh JL. Acceptance and commitment therapy delivered via a mobile phone messaging robot to decrease postoperative opioid use in patients with orthopedic trauma: randomized controlled trial. J Med Internet Res 2020 Jul 29;22(7):e17750
  • Hadlandsmyth K, Dindo LN, Wajid R, Sugg SL, Zimmerman MB, Rakel BA. A single‐session acceptance and commitment therapy intervention among women undergoing surgery for breast cancer: a randomized pilot trial to reduce persistent postsurgical pain. Psychooncology 2019;28(11):2210-2217.

Reviewer 2 Report (Previous Reviewer 3)

This is the second version of a study protocol outlining a study to determine the effectiveness of ACT for the reduction of postsurgical pain. The manuscript has been well improved and the comments of the other reviewers were respected as well. Overall, great job by the authors! This is a scientifically sound protocol and I wish the authors all the best for this interesting study. I only have two minor issues that need to be addressed.

l. 190: It is interesting that the a priori expected effect size changed from the last to the recent version of the manuscript. Does this effect size of Cohen’s d = 0.30 originate from the cited studies? If so, this should be made clear in the according paragraph by citing the exact study here. Again, I need to ask the authors to determine how sample size was exactly calculated. There are many ways to do this in R. Which package did you use to determine sample size? This needs to be as exact as possible to understand your proceedings and make the sample size calculation reproducible.

Regarding the discussion of potential biases: That is exactly how I view it as well (regarding potential biases of the design testing TAU vs ACT + TAU and not TAU vs. ACT). This should be included in the discussion of your manuscript under “3. Discussion”

Author Response

This is the second version of a study protocol outlining a study to determine the effectiveness of ACT for the reduction of postsurgical pain. The manuscript has been well improved and the comments of the other reviewers were respected as well. Overall, great job by the authors! This is a scientifically sound protocol and I wish the authors all the best for this interesting study. I only have two minor issues that need to be addressed.

  1. 190: It is interesting that the a prioriexpected effect size changed from the last to the recent version of the manuscript. Does this effect size of Cohen’s d= 0.30 originate from the cited studies? If so, this should be made clear in the according paragraph by citing the exact study here. Again, I need to ask the authors to determine how sample size was exactly calculated. There are many ways to do this in R. Which package did you use to determine sample size? This needs to be as exact as possible to understand your proceedings and make the sample size calculation reproducible.

Thank you for your comment. Again, we have updated the effect size of the primary outcome (pain interference) at post-treatment (d = 0.64) and at follow-up (d = 0.73) considering the results of a recently published videoconference-delivered group RCT of ACT for people with chronic pain (Sanabria-Mazo et al., 2023). Consistent with the existing literature, we defined pain interference as the primary outcome and pain intensity as the secondary outcome (both measured with the BPI). Moreover, we have decided to adjust the dropout parameters in this RCT to 30% (more realistic scenario). This amendment is based on the results obtained in the aforementioned RCT (Sanabria-Mazo et al., 2023) and on the conclusions of a meta-analysis that explored dropouts in ACT (Ong et al., 2018). Following your request, we have included in this paragraph the specific citation of all these studies. These are the references:

  • Sanabria-Mazo, J.P.; Colomer-Carbonell, A.; Borràs, X.; Castaño-Asins, J.R.; McCracken, L.M.; Montero-Marin, J.; Pérez-Aranda, A.; Edo, S.; Sanz, A.; Feliu-Soler, A.; Luciano, J.V. Efficacy of videoconference group Acceptance and Commitment Therapy (ACT) and Behavioral Activation Therapy for Depression (BATD) for chronic low back pain (CLBP) plus comorbid depressive symptoms: A randomized controlled trial (IMPACT study). J Pain. 2023, S1526-5900(23), 00400-5.
  • Ong, C.W.; Lee, E.B.; Twohig, M.P. A meta-analysis of dropout rates in acceptance and commitment therapy. Behav Res Ther. 2018, 104, 14-33.

To calculate the size, we used the R package "pwr" (specifically “pwr.t.test”). These were the codes we used to estimate the lower and upper limits of the sample size:

  • > pwr.t.test(d=0.64,power=0.80,sig.level=0.05,type="two.sample")
  • > pwr.t.test(d=0.73,power=0.80,sig.level=0.05,type="two.sample")

The results of this calculation yielded a sample size range between 30 (lower limit) and 39 (upper limit) participants per arm. We decided to select the conservative option (n= 39),. Estimating a 30% dropout rate, the minimum number of participants needed per arm will be 51. Therefore, we have increased the initially proposed total sample size of our RCT to 102 participants. This information has been inserted in the main text and in Figure 1.

We would like to thank you for your comments on the sample size calculation of this RCT, as they have helped us to reconsider some of our initial study parameters, which were slightly optimistic.

Regarding the discussion of potential biases: That is exactly how I view it as well (regarding potential biases of the design testing TAU vs ACT + TAU and not TAU vs. ACT). This should be included in the discussion of your manuscript under “3. Discussion”

We agree that it is important to include this information in the "Discussion" section. For this reason, we have added the following two paragraphs at the end of this section:

“Finally, we would like to mention that including TAU alongside an intervention in a clinical trial is not only important from a scientific perspective, but also from an ethical one. It is necessary to consider the patient's perspective and ensure that they receive the appropriate treatment and the best possible care. In some cases, TAU may be the best available treatment for a given condition. In addition, this helps to prevent patients in the clinical trial from being subjected to unnecessary risk by receiving only the experimental intervention without having the standard treatment option.

Nevertheless, including TAU alongside an ACT in an RCT may introduce different types of biases that could affect the interpretation of the results. Confounding bias can make it difficult to separate the effects of treatment as usual from the effects of the experimental treatment. To minimize the biases that may arise when including TAU alongside an intervention in our RTC, carefully monitor variables that could explain the study results, such as patient characteristics (medical history, comorbidities, and medication) and intervention characteristics (adherence to treatment as usual, dropouts, administration format, therapist experience, and patient expectations). Monitoring these variables (which are collected in the RCT battery of measures) can help us to improve the validity of the study results.”

This manuscript is a resubmission of an earlier submission. The following is a list of the peer review reports and author responses from that submission.

Round 1

Reviewer 1 Report

Dear Authors,

I have carefully reviewed your manuscript entitled "Effectiveness of Acceptance and Commitment Therapy (ACT) for the Management of Postsurgical Pain: Study Protocol of a Randomized Controlled Trial (SPINE-ACT study)" and I must say that it is a well-written and comprehensive study. The methodology you have described is particularly commendable and ensures high reproducibility. I would like to make two minor comments to improve the quality of the manuscript.

1. please ensure that you avoid repeating numbers in the references.

2. I would suggest that you reconsider your decision to exclude patients without risk factors for chronic post-surgical pain from the study. Including such patients and then stratifying the results accordingly could provide valuable insights into the effectiveness of ACT in different patient populations. This would also increase the generalizability of your findings and make your study more clinically relevant.

Reviewer 2 Report

Overview and general recommendation:

Thank you for the opportunity to review this study protocol. The authors described the study design of an RCT comparing the effectiveness of perioperative ACT+TAU vs. TAU for spine surgery candidates. Overall, the study protocol is organized and well-designed, and addresses an important research area—non-pharmacological intervention to prevent chronic post-surgical pain and improve postoperative outcomes. I have provided some questions for clarification and feedback for improvement below.

Overall: Please proofread for grammatical and spelling errors and awkward phrasing throughout the paper (e.g., in the Abstract: “…for patients candidates for surgery…”, in the intro “…such as spinal stenosis, facet arthrosis, lead to chronic pain…”, “…benefits of surgery compared to non-surgical…”, “comparint”, etc.).   

Introduction:

1.     Edit for vague wording (e.g., “Low back pain is a very prevalent condition that several people experience…”— provide specific stats).

2.     For readers who are not familiar with psychological intervention or jargon, it would be helpful to briefly describe concepts when you introduce them (e.g., ACT, pain catastrophizing).

3.     Hypotheses are missing from the introduction. In the abstract, the authors predict that ACT will help patients “cope with the stress and uncertainty of surgery and their condition.” However, in that case, why is the BPI (pain interference and intensity) the primary outcome measure? It is unclear whether pain interference is the primary outcome or both pain interference and intensity (keep consistent throughout). Also missing from the introduction is discussion of theory and/or empirical evidence to support hypotheses regarding the benefits of ACT in this population and rationale for chosen mediators.    

4.     The authors mention both mediators and moderators in the introduction (and discussion), yet only describe mediation analysis in the statistical analysis section.

Methods:

1.     Part of the inclusion criteria is that patients present with “psychosocial risk factors for chronic postsurgical pain…” Please specify. Which risk factors? How will this be determined (e.g., cut-offs, etc.)?

2.     Have the authors considered excluding patients who have recently received psychological treatment, particularly ACT? If not, why?

3.     Who will be delivering the ACT intervention?

4.     It is not clear when the ACT intervention will be delivered relative to patients’ surgery (pre, post, combination, etc.).

5.     Given challenges of delivering psychological intervention during the perioperative period (e.g., logistical constraints such as timing of surgery; Pester BD, Edwards RR, Martel MO, Gilligan CJ, Meints SM. Mind-body approaches for reducing the need for post-operative opioids: Evidence and opportunities. J Clin Anesth Intensive Care. 2022;3(1):1-5), have the authors considered using a briefer intervention (e.g., a 1-2 session workshop vs. 8 90-minute sessions) or other methods of maximizing flexibility for surgical patients?

6.     The description of ACT (including table 1) seems quite content-heavy and didactic, rather than experiential.

7.     Proofread for grammatical errors in the statistical analysis section.

Discussion:

1.     There is limited discussion on prior work in this area (i.e., ACT for spine surgery / surgery patients). This section would also benefit from further discussion on the clinical implications of this research.

Reviewer 3 Report

I am excited to review the present manuscript. The authors present the protocol for a study to determine the effectiveness of Acceptance and Commitment Therapy (ACT) in comparison to treatment as usual (TAU) for patients’ postsurgical pain with degenerative lumbar pathology and corresponding surgery. The manuscript is well structured and the research hypotheses are clearly laid out, based on a thorough overview over the respective literature. However, there are some (partially serious) methodological issues I would like to highlight, which may be resolved to further improve this high quality protocol and, hence, the study as a whole.

Section 1: Are there any other non-ACT treatments for this kind of intervention? What effects can be expected, e.g., from a CBT pain management approach? If there are other established approaches already, why does ACT need to be established as well? The literature on this matter is needed to compare this kind of intervention with others and ultimately guide policy makers on which kind of treatment should be supported for patients.

ll. 107 – 113: How was power exactly calculated? This is a very important point in frequentist null-hypothesis significance testing, and that’s why a detailed power calculation is crucial here. Only with a sound power calculation, the sampling of participants makes sense because we do not want to have sample too small (which results in an underpowered study) or too many participants (which makes a significant test statistic more probable even in the absence of an effect and, thus, can be seen as p-hacking). My main question here is: What software did you use and where does d = 0.27 come from? Does this a priori effect size have any basis? Other studies or even meta-analyses regarding ACT and pain should be considered here. If those don’t exist yet, effect sizes for other approaches might be considered.

Furthermore, a 15% dropout seems rather optimistic to me and my experience with clinical trials, especially for the long-term follow-up after 12 months. Maybe you could study the corresponding literature of similar trials. If the number of dropouts is believed to be very low, then less patients would be sampled and the result would be an underpowered study.

ll. 119 – 121: I really appreciate your detailed description of inclusion and exclusion criteria. But at this section in the manuscript one may wonder, what those ICD criteria are and how they are checked. And especially by whom? Do you use any checklists to ensure the consistent recording of diagnostic criteria? You detail some of this information in Section 2.5.1 but I do think you can introduce it here. To my mind, this would improve the flow of the manuscript. But please advise me otherwise, if you don’t think so.

ll. 121- 122: What are psychological risk factors for chronic post-surgical pain? Can you give examples and cite the relevant literature for the interested reader?

ll. 125 – 126: This is an important point and I can fully understand why you want to check this, but I do think that “cognitive impairment” needs to be defined a priori. What exactly do you mean by this? And does the clinician who checks this then have the competence to assess it? Against the ethical background of this study, you should minimize the risk of both, false-negatives and false-positives.

ll. 129 – 131: Who assesses these severe psychological disorders and how? Again, to my mind, you can already describe here who assesses these disorders to improve the reading flow of your manuscript.

ll. 162 – 163: I may be a little picky here, but as far as I understand, ACT is a form of Cognitive Behavior Therapy (third wave CBT). Maybe you should add this phrase to further distinguish ACT from early behavioral approaches for the novel reader.

Section 2.4.1: This section is crucial for your study as well as potential replication attempts and the implementation of your intervention in clinical practice. After reading this section, there remain only a few questions unanswered: Who delivers the treatment? What is the clinician’s qualification? And how is adherence to the treatment manual checked?

Additionally, I appreciate your thorough description of the ACT intervention but at the same time, I do think it can be much improved. Please consider this article regarding the description of treatments for your manuscript: https://doi.org/10.1136/bmj.g1687

ll. 221 – 227: You describe a thorough test battery with reasonable tests. I wonder why you do not employ the BDI and BAI to measure depressive syndrome and anxiety? These are established instruments and are likely used in other intervention studies. This would make results directly comparable to other studies. Please advise me on why you chose those instruments and how well they perform in other studies.

ll. 276 – 277: This is just a small issue and barely worth a mentioning but still important. You most likely mean “R with RStudio”. R is the underlying programming language and RStudio is the development environment in which you use R. Please bear in mind that all used packages will need to be cited with exact version numbers in your final manuscript.

ll. 276 – 283: You do not need to compare your intervention with TAU group regarding their descriptives. I know that this is done frequently, but you may omit the test of statistical significance of two groups after randomization. Roughly speaking, you test if the observed difference happened by chance or not. We already know that any difference between these two groups happened by chance because of your randomization. Thus, your thoughtful test is “not necessarily wrong, just illogical” (CONSORT 2010 Explanation and Elaboration: updated guidelines for reporting parallel group randomised trials, p. 17).

ll. 294 ff.: Please feel free to correct me if I did not understand your research question fully, but what you first describe seems to be a moderation and not a mediation analysis. If I understand correctly, you want to show if change in primary and secondary outcome scores over time is affected by the group membership (TAU vs. TAU + ACT) for different levels of CPAQ and PIPS difference of your patients. You hypothesize that the change in one variable is associated with the change in another. Thus, you would expect an interaction between CPAQ/PIPS-difference, group membership, and time, which is a classic moderation effect. By calculating two difference scores and simply correlating them (and I can see why you did this because it may be a viable option in some circumstances), you lose very much information from your data. If, for instance, a patient enters with a pain score pre intervention = 8 and ends post intervention = 5, and another patient enters with 3 and ends with 0, these two patients have the same difference score but at vastly different pain levels! This kind of information, however, may still be needed and even compulsory for your clinical study. This moderation analysis can easily be implemented in R and multilevel linear models, which you already use. If you use the lme4 package, your function call would be something like “lmer(outcome ~ time * group * cpaq_difference + (1 | participant), data = dataset)”. You will find the parameters package and the modelbased package from the easystats project helpful for this kind of analyses.

My main critique considers the use of your proposed mediation analyses per path analyses. I know that this is done frequently in different settings but this kind of analysis does not yield the results usually described in the respective articles. I strongly recommend the excellent article by Bullock & Green (https://doi.org/10.1177/25152459211047227) which, in line with the majority of methodologists, clearly explains the fact that your mediation effect of interest is always positively biased if it is affected by an unobserved variable that also affects your outcome of interest. In your scenario, one can think of many variables that could influence both, the mediator and your outcome (literacy skills, intelligence, ethnicity, age, gender, prior treatment experience, attitude towards ACT, therapeutic relationship, etc.). Thus, it is highly likely that your mediation effect is positively biased. Furthermore, in such a case, the mediation effect may seem to be present as is indicated by a significant effect, even if there is in fact no mediation at all. One cannot discern a true mediation from a positively biased one. Thusly, the use of mediation analysis as is outlined in your protocol is per se not suitable for this kind of research question. I know that this might be frustrating for you, but this approach would probably lead to “junk science” results.

Section 3: This is a thorough discussion and I follow it. However, there seems to be one important point that needs to be highlighted: You do not compare ACT vs. TAU but ACT + TAU vs. TAU. I wonder if you could elaborate on potential biases, which may be introduced by this procedure. I can fully understand why you chose this design and I myself have conducted such studies before. It isn’t a bad methodology, especially against the background of ethical considerations. But the additional time and relationship patients receive or other factors may introduce a bias that needs to be considered and discussed.

Section 2.: As far as I understand, and please correct me if I am wrong, you solely focus on statistical significance to decide if this additional ACT intervention is “effective” or not. In clinical research, we usually consider statistical and clinical significance as well. (Very) roughly speaking, with statistical significance, you determine if an observed effect would very surprising if there was actually no effect at all. This is a question of probability which excludes the meaningfulness or size of effects. I know that you plan to calculate effect sizes, which is great! But these are based on group summary statistics and may be hard to interpret (see Funder & Ozer, https://doi.org/10.1177/2515245919847202). An extreme example would be an intervention that was administered to 40 patients. On the relevant instrument score, 20 showed a reduction of 15 points And the other half demonstrated an increase in 15 points. In this case, your effect size would be d = 0 but a lot has happened which cannot be seen by calculating effect sizes.

There do exist methods to decide if an intervention has a meaningful effect, relevant to the individual patient. For this kind of question, that may be more important for clinicians, patients, and policy makers, the R package clinicalsignificance may be of interest for you which employs the methods proposed by Jacobson & Truax (https://doi.org/10.1037//0022-006x.59.1.12). Please consider employing this kind of analysis. To my mind, a thorough intervention study which focuses on the intervention effects of a novel intervention is practically more relevant than a biased mediation analysis. If you do not choose to conduct a clinical significance analysis, please advise me why you chose not to do so by citing the relevant literature supporting your argument .